# Microbial Profiling of Potato-Associated Rhizosphere Bacteria under Bacteriophage Therapy

**DOI:** 10.3390/antibiotics11081117

**Published:** 2022-08-18

**Authors:** Samar Mousa, Mahmoud Magdy, Dongyan Xiong, Raphael Nyaruabaa, Samah Mohamed Rizk, Junping Yu, Hongping Wei

**Affiliations:** 1CAS Key Laboratory of Special Pathogens and Biosafety, Center for Biosafety Mega-Science, Wuhan Institute of Virology, Chinese Academy of Sciences, Wuhan 430071, China; 2International College, University of Chinese Academy of Sciences, Beijing 101408, China; 3Agricultural Botany Department, Faculty of Agriculture, Suez Canal University, Ismailia 41522, Egypt; 4Genetics Department, Faculty of Agriculture, Ain Shams University, Cairo 11241, Egypt

**Keywords:** bacteriophage treatments, rhizosphere microbiota, *Solanum tuberosum*, single phage therapy, phage cocktail therapy

## Abstract

Potato soft rot and wilt are economically problematic diseases due to the lack of effective bactericides. Bacteriophages have been studied as a novel and environment-friendly alternative to control plant diseases. However, few experiments have been conducted to study the changes in plants and soil microbiomes after bacteriophage therapy. In this study, rhizosphere microbiomes were examined after potatoes were separately infected with three bacteria (*Ralstonia solanacearum*, *Pectobacterium carotovorum*, *Pectobacterium atrosepticum*) and subsequently treated with a single phage or a phage cocktail consisting of three phages each. Results showed that using the phage cocktails had better efficacy in reducing the disease incidence and disease symptoms’ levels when compared to the application of a single phage under greenhouse conditions. At the same time, the rhizosphere microbiota in the soil was affected by the changes in micro-organisms’ richness and counts. In conclusion, the explicit phage mixers have the potential to control plant pathogenic bacteria and cause changes in the rhizosphere bacteria, but not affect the beneficial rhizosphere microbes.

## 1. Introduction

Potatoes (*Solanum tuberosum* L.) are considered to be the third most consumed crop globally and the main food for more than one billion people in the world [1,2,3]. This means that potatoes contribute significantly to the global food security and economy when used as cash crops [4]. Among these threats, potato infection by bacterial diseases is serious as it may lead to a tremendous crop loss of up to 80% [5,6]. Two major forms of potato bacterial disease exist including potato soft rot and wilt [7]. The potato soft rot is caused by a range of bacteria, including *Pectobacterium carotovorum*, *Pectobacterium atrosepticum*, and *Dickeya* spp., while potato wilt is caused by *Ralstonia solanacearum* [8,9,10]. These bacterial pathogens are soilborne and can infect plants during growth, causing severe damage [11]. Therefore, effective and environmentally friendly control agents can be used to combat these diseases and their associated bacterial pathogens [12,13,14].

Several strategies, including the use of bactericides [15,16,17,18], antimicrobials [16,19], and bacterial inoculants, have been adopted to control potato soft rot and wilt [17,18]. Despite their efficacy, each of these methods has its own demerits [19]. For example, bactericides such as copper compounds, 5-nitro-8-hydroxyquinoline, chlorine dioxide, and mercuric chloride can cause environmental pollution, increase resistant bacterial strains and heighten the price of agricultural production [5,17,19]. Additionally, the use of antimicrobials such as oxolinic acid, streptomycin, and validamycin A for controlling bacteria that can cause soft rot and wilt can lead to resistant strains that ultimately contribute to the already alarming list of antimicrobial-resistant strains [6,20]. Biocontrol using bacterial inoculants to modify the composition of plant rhizosphere microbiota has been proposed as an alternative to pesticides for pathogen elimination [21,22,23]. However, bacterial inoculants are often ineffective owing to their poor establishment in the rhizosphere, competition with native microbiota for resources, and interference with native microbiota [20,24]. As a result, new approaches, including the use of bacteriophages as potential biocontrol agents, are being explored [25].

Bacteriophages (phages) are viruses that infect and propagate within bacterial cells [13,26]. The growing interest in applying phages in the biocontrol of plant pathogens stems from their advantages, including host specificity, environmental friendliness, self-replication, non-toxicity, ability to overcome antimicrobial resistance, cost-effectiveness, ease of production, and the ability to be used as cocktails to improve their efficacy [12,14,25]. Owing to these advantages and more, studies have shown that phages can be used to control soft-rot Enterobacteriaceae (SRE) and potato wilt with satisfactory accomplishment in field trials [10,11,20]. Despite this, experimental evidence on the effects of phages on the native rhizosphere, as well as on the properties of the soil such as pH and organic contents, is still scarce. Additionally, phages can be used as single variants or as cocktails to improve their efficacy. The use of cocktails may further have an additional effect on soil properties and native rhizosphere microbiota [27,28,29].

Recent advancements in molecular diagnostic tools such as sequencing, metagenomics, and bioinformatics can be used to answer these questions [30,31,32]. Using these tools, studies can be conducted to determine how evolutionary trade-offs or phage-mediated pathogen density reduction may affect the composition and functions of the native rhizosphere microbiome [27,29]. For example, a decrease in pathogen density of one bacterium mediated by phages may result in an increased competition of niche space and nutrient uptake by other native bacteria, consequently leading to changes in native rhizosphere and microorganism diversity [29,30]. These changes may have beneficial secondary effects on the plant owing to a reduction in bacterial loads associated with plant diseases [27].

Therefore, in this study, we determined the effects of phage therapy on potato bacterial diseases using three pathogenic bacteria, *R. solanacearum*, *P. carotovorum*, and *P. atrosepticum*. Using greenhouse experiments and metagenomic analysis, we assessed the effects of single and cocktail phages against potato bacterial soft rot and wilt in complex microbial communities and tested whether these effects extend to other microbes within the rhizosphere area.

## 2. Results

### 2.1. Efficacy of Bacteriophage Therapy on Potato Bacterial Diseases In Vivo

The phages used in this study were previously isolated and used to assess their biocontrol efficacy on potato infecting phytobacteria in vitro [1,5]. In this study, we designed a greenhouse experiment using the same phages as single and cocktails (Appendix A) to determine if the phages can also control potato bacterial disease in vivo (Appendix A).

Three bacteria, *R. solanacearum* (Rs), *P. carotovorum* (Pc), and *P. atrosepticum* (Pa), were inoculated to cause potato wilt and soft rot diseases, respectively. As shown in Figure 1, all of the five plants in the positive control group (inoculated with bacteria Rs, Pc, or Pa, without phage treatment) showed signs of bacterial infection (the ratios were 0:5 in terms of healthy to infected plants for Rs, Pc, and Pa). The single phage treatments showed differences in disease incidence, with ratios of 4:1, 5:0, and 3:2 in terms of healthy to infected plants for SRs, SPc, and SPa, respectively. On the other hand, the phage cocktail treatments were more effective for the reduction of the diseases’ incidence, with ratios of 5:0, 5:0, and 4:1 in terms of healthy to infected plants for RsPck, PcPck, and PaPck, respectively (Figure 1A).

Notably, the phage cocktails treatments showed a remarkable plant growth than all groups indicating that it may have killed the three bacteria causing potato wilt and soft rot diseases. In detail, the percentage of disease symptoms revealed in Rs-treated plants was 80%, the percentage ranged between 10–20% when the single-phage (SRs) was applied, and decreased down to 0–5% when the phage cocktail (RsPck) was used. For the Pc-treated plants, the percentage of disease symptoms was 70% and the percentage of disease symptoms revealed after inoculation with SPc ranged between 10–25% and decreased down to 0–7% when the PcPck treatment was applied. The percentages of disease symptoms caused by Pa was about 70%, decreased down to 25% with the application of SPa and reduced down to 8% when the PaPck treatment was applied. The negative control remained asymptomatic during the experiment period. Reductions were significant (*p*-value < 0.001) in all the applied phage treatments. Data indicated that phage cocktails were more effective than single phage treatments (Figure 1B).

### 2.2. Microbial Communities: Pathogens and Phage Therapy

#### 2.2.1. Rhizosphere Microbiome Profiling

The microbial profiling of the soils yielded average total OTU (Operational Taxonomic Unit) counts of 1294, 1272, and 1280 for the Rs, Pc, and Pa groups, respectively. After phage treatments, the total OTUSs were 1286, 1272, and 1280 for SRs, SPc, and SPa, respectively, and 1285, 1207, and 1291 for RsPck, PcPck, and PaPck, respectively, versus the average total OTU of the native soils which was 794. Among all samples, the common OTU count was 1075, separated into 695 OTUs shared with the native soil sample and 380 OTUs exclusively shared among the treated samples (Figure 2).

The average Shannon index (i.e., an index to measure the diversity of species in a community) for replicates per treatment was applied to estimate the detected diversity within each sample (i.e., alpha diversity). Among all phage therapy treatments, the phage cocktail (PaPck) and the single phage (SPa) had the highest diversity, followed by the single phage (SRs) and phage cocktail (RsPck). The difference was significant with *p* < 0.01. 

The single phage (SPc) and phage cocktail (PcPck) had the lowest diversity compared to native soil sample. The Shannon index ranged from 4.06 to 5.21. In detail, the Shannon diversity index values of Rs, SRs, and RsPck were 5.21, 5.04, and 4.86, respectively, and 4.99, 4.68, and 4.32 for Pc, SPc, and PcPck, respectively, while the values for Pa, Spa, and PaPck were 4.51, 4.25, and 4.28, respectively, when compared to the native soil (4.06).

#### 2.2.2. Rhizosphere Microbial Communities

Proteobacteria were highly abundant among all phage treatments (percentages of 61, 59, 57, 55%, for SRs, RsPck, PcPck, and SPc, respectively) when compared to the negative control (57%). Firmicutes was highly abundant in the phage therapy treatments (31, 25%), for Spa and PaPck, respectively, compared to the negative control (2%). Additionally, Bacteriodota was highly abundant in the phage therapy treatments (19, 17, 10%), for PcPck, SPc and RsPck, respectively, compared to the negative control (7%). In contrast, Actinobacteriota had relatively low abundance in the phage therapy treatments (10, 8, 7%) for RsPck, PcPck, and PaPck, respectively, compared to the negative control (13%; Figure 3A).

The commonly shared OTUs among the phage therapy treatments revealed significant differences in six microbial phyla which included Proteobacteria, Firmicutes, Acidobacteriota, Actinobacteriota, Bacteriodota, and Gemmatimonadota. Among all samples of the three phage therapy groups, the phage therapy group of *R. solanacearum* (Rs) revealed the abundance of highly bacterial phyla, generally being Proteobacteria, followed by Actinobacteriota, Bacteriodota, Gemmatimonadota, Acidobacteriota, Firmicutes, Acidobacteriota, Myxococcota, and Chloroflexi, while the most identified bacterial phyla of the phage therapy group of *P. carotovorum* (Pc) generally was Proteobacteria, followed by Bacteriodota, Actinobacteriota, Firmicutes, Acidobacteriota and Gemmatimonadota. In contrast with the phage therapy group of *P. atrosepticum* (Pa), the most identified bacterial phylum was Proteobacteria, followed by Firmicutes, Actinobacteriota, Bacteriodota, Gemmatimonadota and Acidobacteriota.

Regardless of the phage therapy type, the abundance of Firmicutes was significant in the phage therapy treatments compared to the negative control among all groups. The Actinobacteriota, Bacteriodota and Firmicutes phyla were the most presented among all with almost an equal distribution among different treatments (Figure 3B).

#### 2.2.3. Phage Therapy-Related Microbial Communities

After initial screening, all detected genera (nodes) were retained in two clusters and compared to uncultivated soil. On average, phage cocktail (RsPck, PcPck, and PaPck) networks were more connected and had shorter path lengths. Instead, most of the taxa associations were completely different between the phage cocktail and the three pathogen (Rs, Pc and Pa) communities, and the number of significant associations increased with the number of phages when compared to uncultivated soil (potato) (Figure 4A).

The microbiome composition and diversity at the family level was investigated among the three phage-therapy treatments at the pathogen, single-phage, and phage cocktail treatments, independently (Figure 4B). The bacterial species belonging to Bacillaceae family were common among Pa groups. In comparison, Pseudomonadaceae and Flavobacteriaceae were common among PcPck. In contrast, Sphingomonadaceae, Xanthomonadaceae and Moraxellaceae were shared between all three groups.

#### 2.2.4. Phenotypic Prediction of Phage Treated Groups

Based on the recorded metadata for microbial species in databases, phenotypic categories were defined. The phenotypic profiles of the rhizosphere of phage therapy treatments and negative control were compared and controlled by the phage therapy groups. The rhizosphere microbial community showed a significant difference among the phage treatments. For the first group of the experiment (Rs), the phage cocktail (RsPck) presented a significant difference to the negative control, being highly effective with the pathogen (Rs) among facultatively anaerobic microbes. Moreover, significant differences between RsPck and the negative control with potentially pathogenic microbes (Figure 5A) were found.

The second group (Pc) showed that both single phage (SPc) and phage cocktail (PcPck) had significant difference with the negative control samples among the anaerobic group of bacteria. Furthermore, significant differences between single phages (SPc) and pathogens (Pc) among facultatively anaerobic microbes were found (Figure 5B).

In contrast, the third group (Pa) showed significant differences among the single phage (SPa) treatment and the pathogen (Pa) with the negative control samples at mobile elements, and significant difference between single phage (SPa) and the negative control, being potentially pathogenic (Figure 5C).

#### 2.2.5. Functional Prediction of Phage Treated Groups

The functional properties of the detected bacterial taxa were investigated in relation to the different phage therapy treatments based on KEGG pathways. Based on the enriched pathways values of the negative samples (*x*-axis) in contrast to all the other samples (*y*-axis) the most represented functional pathways were detected (Figure 6). The most represented pathways were related to the organism’s metabolism for all samples, followed by the biosynthesis of secondary metabolites, microbial metabolism in diverse environment, biosynthesis of amino acids and carbon metabolism. In the case of the environmental information processes group, the two-component system and ABC transporters were distinguished, while ribosomes’ formation was highly presented as the genetic information processes group. It was observed that the PaPck was highly represented when compared to the other treatments (Figure 6A). Additionally, the purine metabolism, oxidative phosphorylation, pyruvate metabolism and glyoxylate dicarboxylate metabolism of the metabolism group were presented at lower levels, as well as the quorum sensing of the cellular processes group. Equally, the PaPck was more highly represented than the other treatments, followed by SPa and Pa. The RsPck and PaPck were the only treatments that showed higher enrichment levels of the previous groups when compared to their pathogen, or single phage-treated samples, in contrast to the PcPck, which showed the lowest enrichment pathways among all (Figure 6B).

## 3. Discussion

The impact of microbe–microbe interactions on the host–microbial pathogen interaction outcomes is a relevant subject in microbiology and plant pathology. Studies have shown that the microbiome structure, assemblage, and compositions are directly influenced by soil biotic and abiotic factors [31]. Phage therapy is a common practice that has been previously reported in agriculture and plant protection fields [12,13,14]. However, the commercial use of phages in agriculture is still limited [12,13,14]. This study has shown that phages can be used effectively as biocontrol agents while improving overall plant health. This effectiveness was observed after an in vivo evaluation of different phage treatments following a specific sampling design. It was apparent that all the tested phages were able to control potato wilt bacteria *R. solanacearum* and the soft rot bacteria *P. carotovorum* and *P. atrosepticum* as previously reported [1,5]. Although a single phage decreased the occurrence of bacterial wilt and soft rot diseases in contrast to the control, the occurrence of diseases was reduced more by the phage cocktail that contained three different phages under greenhouse conditions. The decline in occurrence of infections could be clarified by a decline in pathogens densities and this impact became stronger with the use of phage cocktails.

Bacteriophages are known for their specificity to bacteria, thus, the phage therapy should affect its specific host (i.e., pathogen) and show an insignificant effect on the natural rhizosphere microbiota [26]. The clear divergence in both species’ richness and counts in the pathogen-treated samples confirm the association of the pathogen with different microbial groups. Thus, the reduction or elimination of this pathogen by phages would eventually cause differences in the existing rhizosphere microbial community represented in the anaerobic microbes that will contribute to facilitate phosphate solubilization and promote the precipitation of soluble Cd in the soil, as well as the facultative anaerobes capable of reducing Fe (III). Effectively, in the current study, the observed changes in the rhizosphere microbiota confirmed the vital role of phages in shaping the potato-related rhizosphere microbiome. The enhancement of plant health after the application of different types of phages may not be only limited to the elimination of the pathogen but also due to the new shifts in the microbial composition.

For example, the NGS metabarcoding-based microbiome profiling revealed the predominance of the species belonging to the phylum Proteobacteria regardless of the treatment group. The Proteobacteria were previously found to be associated with bioremediation of environmental contaminants and the production of highly beneficial phytohormones such as the indole-3-pyruvate pathway for synthesis of the auxinic phytohormone indole acetic acid (IAA) in Azospirillum and Enterobacter genera which belong to the phylum Proteobacteria [32,33,34,35]. Moreover, Azospirillum, Burkholderia, and other genera have the ability for nitrogen fixation by nitrogenase-encoding genes nifHDK [32,36]. However, Pseudomonas belonging to Proteobacteria can synthesize the pyrroloquinoline quinone-encoding genes pqqBCDEFG that can contribute to mineral phosphate solubilization [37], production of the 1-aminocyclopropane-1-carboxylate (ACC) deaminase gene acdS that enables the degradation of the plant’s ethylene precursor [38,39], and synthesis of antimicrobial compounds by the genes hcnABC (hydrogen cyanide) and phlACBD (2,4-diacetylphloroglucinol) [40].

However, Firmicutes are capable of producing ACC deaminase and suppress pathogens which leads to enhanced plant growth and pathogen suppression [41]. Members of the genus Bacillus, which belongs to the phylum Firmicutes, secrete exopolysaccharides and siderophores that inhibit or stop the movement of toxic ions and help maintain an ionic balance [42]. As well as this, they are the direct synthesis of antimicrobial compounds, phytohormones, and siderophores that inhibit or stop the movement of toxic ions and help maintain an ionic balance [43]. An additional feature of the Bacillus genus is its ability to form biofilms, as the biofilm provides a matrix on which the community can develop [42]. This bacterial genus belongs to the phylum Actinobacteriota which contributes to the rotation of soil components into organic components through the decomposition of a complex combination of organic matter in lifeless plants, and animals, in addition to fungal material [44]. The most abundant genus belonging to Actinobacteriota are Streptomyces, which are a prolific source of antimicrobial, and extracellular enzymes. They have the ability to produce secondary metabolites of biotechnological and clinical importance that can play a major role in nutrient cycling. The Streptomyces importance is revealed as biocontrol agents, plant growth-promoting bacteria, and efficient biofertilizers [45].

Therefore, the phylum Bacteriodota contributes to mineral phosphate solubilization as well as the family Cyclobacteriaceae [46]. The bacterial species of the phylum Acidobacteriota have genes that probably help in survival and competitive colonization of the rhizosphere, leading to the establishment of beneficial relationships with plants, regulation of biogeochemical cycles, decomposition of biopolymers, exopolysaccharide secretion, and plant growth promotion [47]. The species belonging to the phylum Chloroflexi are known as anaerobic microbes that can co-exist with methane-metabolizing microbes and are crucial organic matter degraders under anoxic conditions [48]. Methane metabolism is used for the bioremediation of Cd contamination and promotes the precipitation of soluble Cd in soil [48]. Gemmatimonadota is known as the eighth-most abundant bacterial phylum in soils, representing about 1–2% of the soil bacteria worldwide. They are capable of anoxygenic photosynthesis and are associated with the plants and the rhizosphere, treatment plants, and biofilms [49]. The phylum Myxococcota, is broadly distributed in soil with the ability to produce diverse secondary metabolites acting as antimicrobials, antiparasitic, antivirals, cytotoxins, and anti-blood coagulants [50].

Notably, we found that most of the phyla which are presented correlated positively with the functional prediction. The Proteobacteria and Firmicutes both have the ability to produce ACC deaminase, antimicrobial compounds, and phytohormones, while the phylum Bacteriodota facilitates phosphate solubilization in the soil, as well as the phylum Proteobacteria. Therefore, Actinobacteriota and Myxococcota can produce secondary metabolites of biotechnological and antimicrobials. As well as this, Gemmatimonadota, Firmicutes and Acidobacteriota phyla are known microbes for association with the plants and the rhizosphere, treatment plants and plant growth promotion. These function predictions of the detected bacterial taxa support the hypothesis that phage mixers have the potential to control plant pathogenic bacteria and cause changes in the rhizosphere bacteria but not affect the beneficial rhizosphere microbes.

The limitations of these types of treatments are the application of the phage therapy to the field without studying its effect over many plant generations, which will require more time and effort and to be tested over different climatic conditions and soil types. Our continuous plan to overcome this limitation includes: testing it in the field over different seasons from different cultivation spots; observing the phage biocontrol effect on *R. solanacearum, P. carotovorum,* and *P. atrosepticum* for longer time periods, such as 1 or 2 years; studying the histopathology of plants at the cellular level. Additionally, we are planning to apply a whole genome metagenomic analysis to study the bacteriophage therapy effect on the wider microbiome community, including protists and fungi.

## 4. Materials and Methods

### 4.1. Bacterial Isolates and Culture Conditions

The bacterial phytopathogens used in this study included *Ralstonia solanacearum* GIM1.74 (Rs), *Pectobacterium carotovorum* subsp *carotovorum* KPM17 (Pc), and *Pectobacterium atrosepticum* WHG10001 (Pa). The *R. solanacerum* GIM1.74 (Rs) strain was cultured on CTG agar plates and in broth (1% Casamino acid hydroxylate, 1% tryptone and 1.5% *w*/*v*, agar) at 28 °C with shaking (170 r.p.m.). The bacterial species of the genus *Pectobacterium* were cultured on Luria Bertani (LB) agar plates (1.5% *w*/*v*, agar) and in broth at 28 °C [51]. After the incubation, the bacterial culture count in the suspensions ranged between 10^7^ to 10^8^ CFU/mL.

### 4.2. Phage Isolates, Amplification, and Tittering Conditions

Three phages, PSG11, WC4, and CX5, that were previously reported as specific bacteriophages for *R. solanacearum*, *P. carotovorum*, and *P. atrosepticum*, respectively, were used as single phages [1,5]. Three bacteriophage cocktails that each included three different types of bacteriophages were prepared individually: the PSG2/PSG3/PSG11, WC1/WC2/WC4 and CX2/CX3/CX5 phage cocktails. A list of the used bacteria and phages is provided in Table 1.

All the used bacteriophages were prepared in Tris-HCl phage buffer at pH 7.5 (50 mM Tris-base, 150 mM NaCl, 10 mM MgCl_2_.6H_2_O and 2 mM CaCl_2_). Purified phages were amplified by mixing 500 µL of the host bacteria with 10 µL of their respective phage. The mixture was vortexed at 160 rpm and incubated at 28 °C for 15 min. Thereafter, 4 mL of soft agar were added to the phage–bacteria mixture, poured on LB and CTG agar plates and incubated overnight at 28 °C. The overlay agar was scrapped off from the double agar plate into a 15 mL centrifuge tube containing 2 mL phage buffer, followed by vertexing for 2 min and centrifugation at 5500 rpm for 15 min at 4 °C. The phage lysate was then filtered through a syringe-driven filter (0.22 µm). The titer of the phages was determined through 10-fold serial dilutions and placing a spot of 10 µL of the lysate on a double agar layer containing the host bacteria.

### 4.3. Greenhouse Experiment Design and Treatments

The efficacy of single and cocktail phages for controlling potato bacterial wilt and soft rot were tested in pots. All experiments were carried out in the greenhouse of the Wuhan Institute of Virology (Wuhan, China) in the period between August to October 2019. The temperature, relative humidity, and light density fluctuated between 28–37 °C, 58–85%, and 15–20 Klux, respectively.

Soil materials that were used in the present study were collected from a field located in the Wuhan Institute of Virology (Zhendian street, Jianxia district, Wuhan, China) at 0–30 cm depth. Then, the soil was air-dried, grinded and sieved through a 2 mm sieve. Some properties of the soil including pH, particle size distribution, soluble cations and anions were determined according to the Olsen method [52] (Appendix A). The nonsterile soil was uniformly packed in plastic pots of 18 cm height and 26.5 cm mean diameter at a rate of 5 kg soil pot^−1^ (with a 1 cm drainage hole). The soil in each pot was mixed with 50 g cattle manure (CM) (1% *w/w*) as an organic fertilizer. Potato seeds were surface sterilized with 3% NaClO for 5 min and followed by 70% ethyl alcohol for 1 min prior to cultivation. The seeds were then germinated on water–agar plates for two days and further transplanted to each pot. A suspension of pathogens (10^7^ cells/gram of soil) were inoculated onto the plants after 4 days from transplanting, while the phage treatments (10^6^ particles/gram of soil) were inoculated 2 days after the pathogen inoculation. All pots received the same P fertilization at a rate of 1.0 g superphosphate (15.5% P_2_O_5_) per pot, an equivalent to 31.0 kg P_2_O_5_ per feed mixed with the soil before cultivation. Thereafter, three potato seeds were sown in each pot and irrigated to about soil field capacity using tap water. After two weeks, the seedlings were thinned to 1 plant per pot. Then, ammonium sulphate and potassium sulphate at rates of 0.60 g N and 0.25 g K_2_O pot^−1^ (equivalent to 120 kg N and 50 kg K_2_O fed^−1^, respectively) were applied to all pots twice with 20 and 50% of the total amounts after 25 and 50 days from sowing date, respectively.

The experiment was designed as a randomized complete block where ten treatments for the three pathogens were categorized as follows: negative control (i.e., potato seeds cultivated in untreated soil), pathogen-treated samples (i.e., potato seeds cultivated in soil treated with specific bacterial pathogen), single phage treatment (i.e., potato seeds cultivated in soil treated with the specific bacterial pathogen and treated with a single bacteriophage species), and phage-cocktail treatment (i.e., potato seeds cultivated in soil treated with the specific bacterial pathogen and treated with three mixed bacteriophage species), compared to native soil. Treatments were replicated five times and rearranged randomly every four days. Each replicate contained one potato plant per pot. The pathogen treatments were named by pathogen initials, as Rs, Pc and Pa. The single phage treatments were prefixed by the letter “S” (i.e., SRs, SPc and SPa), while the phage cocktail treatments were suffixed by “Pck” (i.e., RsPck, PcPck and PaPck; Figure 7).

### 4.4. Metabarcoding Analysis

For every pot, soil samples were collected randomly before the beginning of the experiment and at the end of the greenhouse experiment from the plant–rhizosphere area and kept in plastic bags for determining the changes in the rhizosphere microbiome composition. Samples were then sent to the company Sangon Biotech (Shanghai, China) for DNA extraction and metabarcoding analysis. A total of 50 samples were collected along with one sample from uncultivated soil (i.e., the source of all the soil used to conduct the experiment). DNA extraction was performed using the Power Soil MoBio DNA Isolation Kit (MO BIO Laboratories Inc., Carlsbad, CA, USA) at a final elution volume of 150 mL.

The bacterial communities in the soil were assessed by sequencing amplicons of the V3–V4 variable region of the 16S rRNA gene, with the primer pair 338F (5′-ACT CCT ACG GG AGG CAG CAG-3′) and 806R (5′- 489 GGA CTA CHV GGG TWT CTA AT-3′). The PCR reaction was performed using the TransStart FastPfu DNA Polymerase mixture. The reaction mixture (20 μL) was composed of 4 µL of 5x FastPfu Buffer, 2 µL of 2.5 mM (each) dNTPs, 0.8 µL of 5 µM Bar-PCR primer F, 0.8 µL of 5 µM primer R, 0.4 µL of FastPfu polymerase, 0.2 µL of BSA and 10 ng of template DNA. Amplification conditions for PCR were as follows: 3 min at 98 °C to denature the DNA, followed by 27 cycles of denaturation at 98 °C for 10 s, primer annealing at 60 °C for 30 s, and strand extension at 72 °C 45 s, followed by 7 min at 72 °C on an ABI Gene Amp 9700 thermocycler (IET, London, UK). Electrophoresis on 2% agarose gel was used to check the quality of the PCR products and purified using Agencourt AMPure XP beads (Beckman, Brea, CA, USA). The pooled DNA product was used to construct an Illumina pair-end library followed by Illumina-adapter ligation and sequencing by Illumina (MiSeq, PE 2 × 300 bp mode), following the manufacturer’s instructions.

Paired-end data were demultiplexed into each sample based on the index sequences downloaded from the Illumina MiSeq platform. Hence, the paired-end sequences of each sample were trimmed based on their quality and length using Trimmomatic [53] and FLASH [54] software. The metabarcoding analysis was performed using the online Majorbio Cloud Platform (http://en.majorbio.com/ (accessed on 1 June 2022)). Uparse V7.1 (http://drive5.com/uparse/ (accessed on 1 June 2022)) was used to detect and remove chimera sequences. Mothur v.1.9.0 software [55] was used to infer richness and to perform library comparisons. The Operational Taxonomic Unit OTU (is the basic unit in numerical taxonomy and can be used to classify groups of closely related species, individuals, or genes) was clustered at a sequence similarity of 0.99, while the taxonomy was identified by the RDP classifier algorithm (http://rdp.cme.msu.edu/ (accessed on 1 June 2022)) versus the Silva 16S rRNA database (version 138) at a 70% confidence threshold. The PICRUSt (http://huttenhower.sph.harvard.edu/galaxy/ (accessed on 1 June 2022)) was employed to predict the functional characters of the detected microbial communities and functions. The co-occurrence network was analyzed using Orange V3.24.1 (https://orange.biolab.si/ (accessed on 1 June 2022)). The counts were analyzed and visualized using Venn diagrams (vegan R-package) and Circos plots (Circos -0.6; http://circos.ca/ (accessed on 1 June 2022)).

## Figures and Tables

**Figure 1 antibiotics-11-01117-f001:**
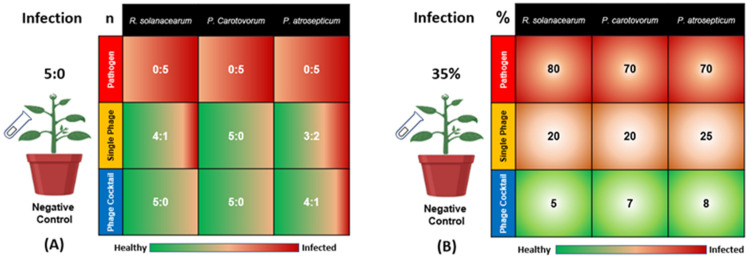
Effects of phage therapy on the incidence of potato bacterial disease in terms of healthy: infected plants for all treatments (**A**) and Percentage of disease symptoms after different treatments (**B**). Positive control or pathogen groups: inoculated with *R. solanacearum* (Rs), *P. carotovorum* (Pc), or *P. atrosepticum* (Pa), respectively; Single phage groups (SRs, SPc, and SPa): inoculated with *R. solanacearum* (Rs), *P. carotovorum* (Pc) or *P. atrosepticum* (Pa) and then treated with a single phage, respectively; Phage cocktail groups (RsPck, PcPck, and PaPck)**:** inoculated with *R. solanacearum* (Rs), *P. carotovorum* (Pc) or *P. atrosepticum* (Pa) and then treated with phage cocktails, respectively. “n” is representing the number of healthy: infected plants and “%” is indicating the percentage of the disease symptoms revealed on the infected plants.

**Figure 2 antibiotics-11-01117-f002:**
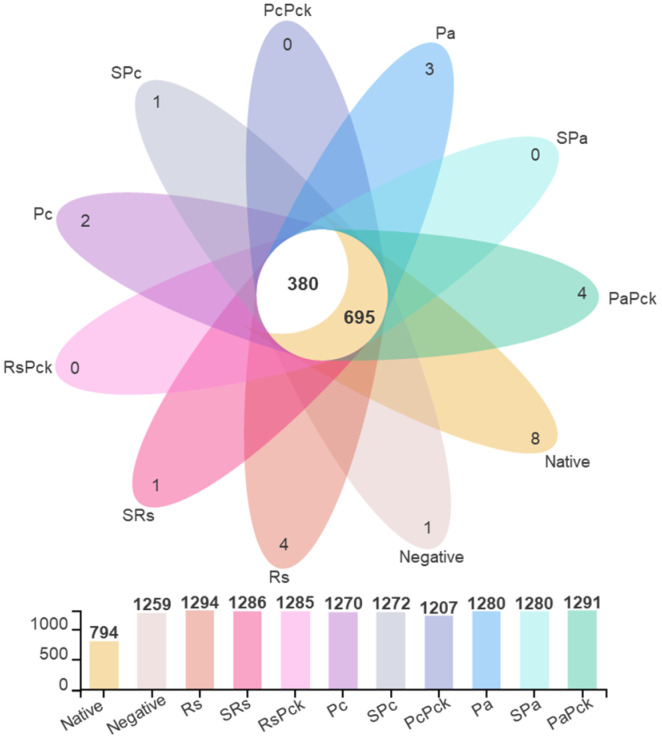
The OTU diversity of three phage therapy groups, positive control, or pathogens group (*R. solanacearum* (Rs), *P. carotovorum* (Pc), and *P. atrosepticum* (Pa)), single phage group (SRs, SPc, and SPa) and phage cocktail groups (RsPck, PcPck, and PaPck) compared to the negative control (no treatments added) and native soil samples, represented by different colors.

**Figure 3 antibiotics-11-01117-f003:**
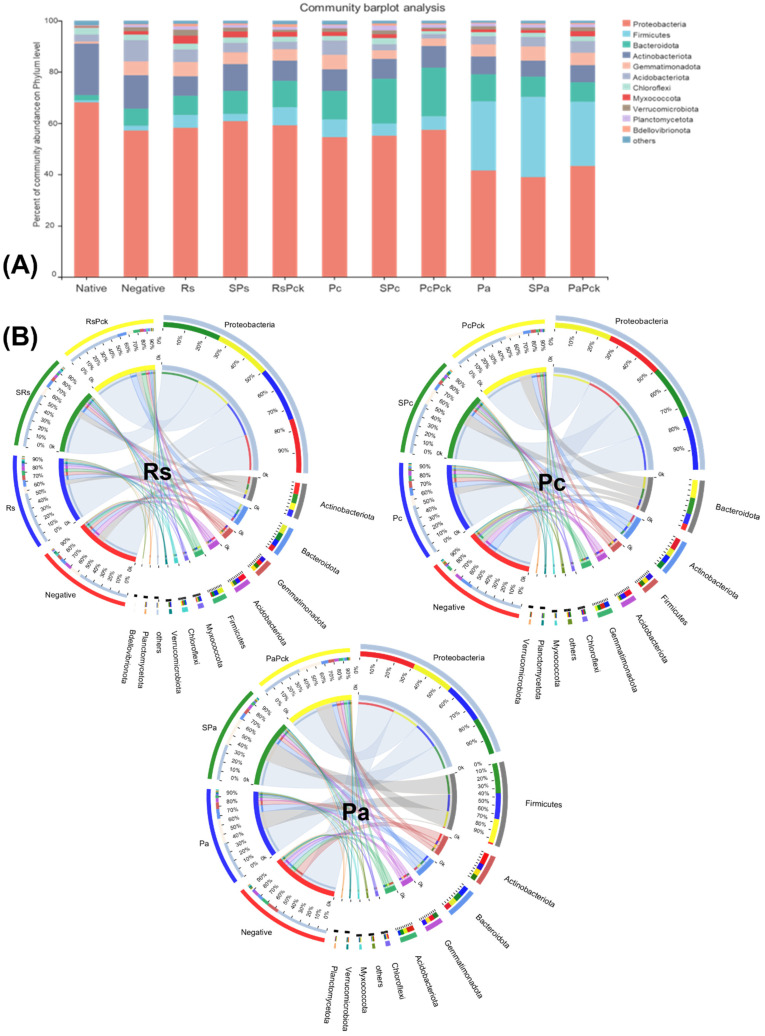
The microbial composition of the surveyed microbiota at phyla level, presented as (**A**) barplot and (**B**) comparative circus plot for each of the three phage therapy treatments (Rs, Pc, and Pa).

**Figure 4 antibiotics-11-01117-f004:**
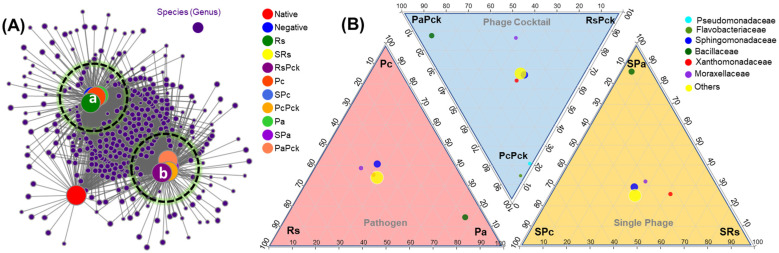
(**A**) Bacterial co-occurrence networks between single, phage cocktail and pathogens for the three phage therapy experiment (a: cluster of the negative, and pathogen treated samples, and b: cluster of phage treated samples) compared to an uncultivated soil sample (native). (**B**) Three triangular comparisons at the family level among the different pathogens (Rs, Pc and Pa), phage cocktails (RsPck, PcPck and PaPck), and single phages (SRs, SPc and SPa) are shown.

**Figure 5 antibiotics-11-01117-f005:**
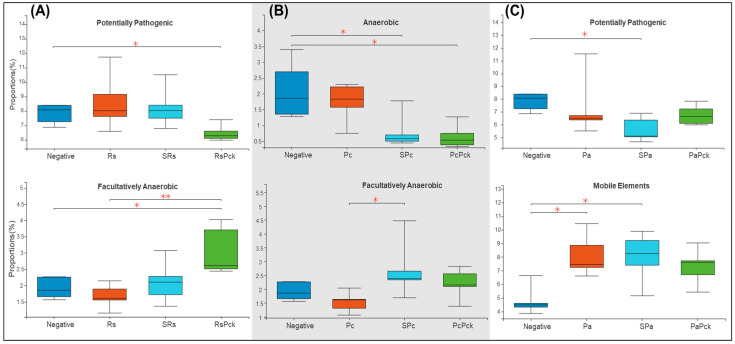
Boxplot diversity explained by phenotypic prediction among the phage therapy treatments. (**A**) Proportions of the potentially pathogenic and facultatively anaerobic phenotype of *R. solanacearum* phage therapy (Rs) treatments. (**B**) Proportions of anaerobic and facultatively anaerobic phenotypes of *P. carotovorum* phage therapy. (**C**) Proportions of the potentially pathogenic and facultatively anaerobic phenotype of *P. atrosepticum* phage therapy (* *p*-value < 0.05, ** *p*-value < 0.001).

**Figure 6 antibiotics-11-01117-f006:**
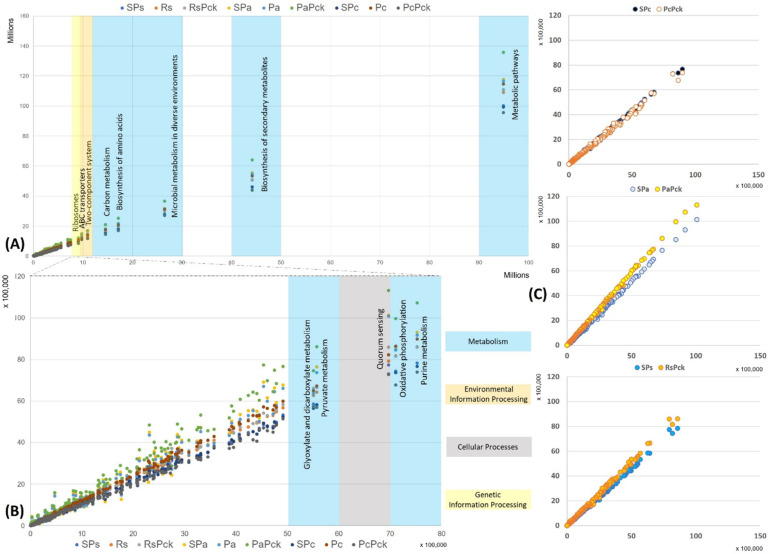
The functional prediction KEGG pathways for all samples at three levels, (**A**) top pathway above 10 million enriched genes, (**B**) between 5500K to 8000K enriched genes and (**C**) the gene enrichment plot of the phage therapy treatments versus the untreated soil.

**Figure 7 antibiotics-11-01117-f007:**
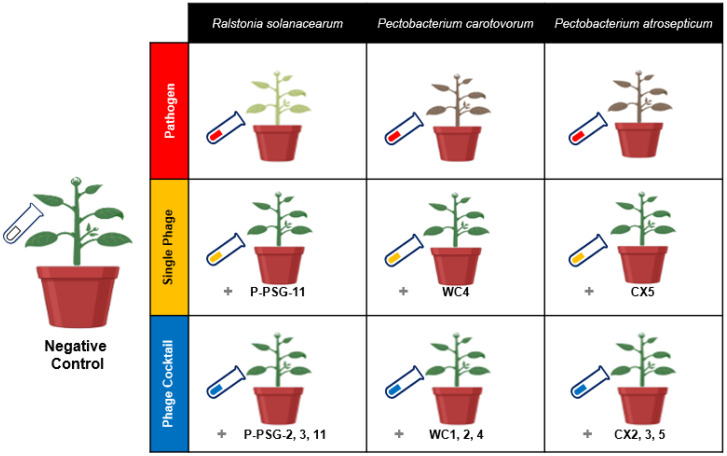
Illustration model for the biocontrol of the three groups, pathogens group *Ralstonia solanacearum* (Rs), *Pectobacterium carotovorum* (Pc) and *Pectobacterium atrosepticum* (Pa), single phage group SRs (PSG11), SPc (WC4) and SPa (CX5) and phage cocktail groups RsPck (P-PSG-2, 3, 11), PcPck (WC1, 2, 4) and PaPck (CX2, 3, 5), compared with negative control (no bacteria or phage added) in a greenhouse experiment.

**Table 1 antibiotics-11-01117-t001:** Bacterial isolates and sources and well as phages are shown.

Bacteria		Phage	
Strain	Source	Single	Cocktail
*Ralstonia solanacearum* strain GIM1.74 (Rs)	Purchased from Guangdong Microbiology Culture Center, China	P-PSG-11 (SRs)	Rs 2, 3,11 cocktail (RsPck)
*Pectobacterium carotovorum* subsp *carotovorum* strain KPM17 (Pc)	Isolated from Molo, Kenya	Wc4 (SPc)	Wc1, 2, 4 cocktail (PcPck)
*Pectobacterium atrosepticum* strain WGH10001 (Pa)	Isolated from Mongolia, China	CX5 (SPa)	CX 2, 3, 5 cocktail (PaPck)

## Data Availability

The raw data is available at NCBI BioProject: PRJNA867554.

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
