# Peer review of "Microbial Profiling of Potato-Associated Rhizosphere Bacteria under Bacteriophage Therapy"

_antibiotics, 2022, doi:10.3390/antibiotics11081117_

Round 1

Reviewer 1 Report

The paper from Mousa et al is a study of the microbial profiling of potato-associated rhizosphere bacteria under bacteriophage control. It falls within the scope of the journal's bacteriophages section.  

Overall impression is that the manuscript is very difficult to follow for a non-specialist reader without more information provided by the authors on their experimental approaches, the rationales for these approaches and better presentation of their data.

To go through the paper by sections:

1. Introduction - generally fine,

a) though Dickeya is the correct spelling (line 36) 

b) line 33, should be 'serious' not 'vital'

c) line 47 - what antibiotics are being used? List them

2. Materials and Methods - this should really follow the introduction, so we know what the authors are going to do, and provides an opportunity to explain acronyms, etc.

a) line 374. You add cattle manure to the soil samples, but will this not add its own microbiota? Even after surface sterilization? This should be discussed - is it a limitation of the study without knowing the microbiota of the added material.

b) section 4.4. the rationale for the metacoding analysis is not explained in the results section. What is an OTU? explain to the lay reader.

3. Results - my main issues are with this section. 

a) the authors should show that the bacteriophages are cidal for the bacteria in vitro, e.g. using plaque assays.

b) the authors should show the different disease pathologies in the controls infected with the bacteria and the abscence of pathology in the phage treated plants to be convincing. Some pictures would be good.

c) Supplementary tables should be included in the main text. 

 d) Line 100 -109 and Figure 1. This is very confusing to the reader. I suggest that the data are presented graphically and not as a coloured picture.

e) line 121 and 124. Again authors need to explain what OTUs are (not OUTs) and line 125, what is a Shannon index? Not enough explanation of the methodology is provided for the lay reader to understand. 

f) line 173 - what is a circos plot? 

g) Figure 6. This figure is a bit difficult to follow.

4. Discussion -generally OK, though not sure of the overall impact of the proposed treatment. Do the bugs have any anti-phage activities, e.g. CRISP-CAS? This should be discussed. Also, the limitations of the study should be discussed.

Reviewer 2 Report

Some additional clarity could be provided in the methodology and associated results section(s) for the greenhouse experiments  - to make it absolutely clear whether the soil, the potato seed or the growing plant is receiving the inocula of pathogen/phage. In some places it would seem that the soil received the pathogen/phage inocula but in other places (including the images) , it suggests that the plant itself was inoculated. If the latter is the case, this leads to another question of why the soil was chosen for metagenomic analyses and not the plant?

Improve the descriptions of experimental design and outcomes section 2.1 - the use of the abbreviations (RS, SRs, RsPck etc) solely is not very accessible  - over-condensed. 

Fig 1 Legend - need to explain n and that data presented represents ratio and ratio of what. Similarly with %.

Lines 133-136 : Could be expanded to detail how this conclusion is reached. Sentence itself is poorly constructed/unclear.

Fig 2 Negative control - what is this? Were differences in diversity significant?

Fig 2B - correction needed to % axis.

Section 2.2 and associated methodology/Figures - greater clarity need on soil sampling, i.e. the 50 soil samples taken in relation to the five replicates section 4.3 and the data presentation in Fig 2.

Fig 3 Figure legend should be provided with appropriate analysis

Fig 4A  What do the and b represent? Need to provide details in figure legend. What does the dashed line represent? 

Some details in legend for Fig 4B - outcome of what analysis is being presented?

Section 2.2.4 and accompanying Figures are difficult to follow.  This is due to lack of clarity due to language combined with lack of detail on the approach used/questions being asked and methodology for answering. The opening sentence refers to phenotypic categories but commentary and Figure only refers to potential pathogens/facultative anaerobes and mobile elements. What is encompassed in each of the three 'categories' is not given and these three categories are limited in differentiation (potential pathogen vs facultative anaerobe)/not of phenotypic relevance (mobile element). There could also be overlap in genera/species membership between these three 'categories'? Details lacking from the Fig legend - for example - meaning of *? are differences significant - determined how? What is the statement 208-210 based on - literature or data generated in this study? What is conclusion 210-213 based on? Where is the analysis of 'beneficial rhizosphere microbes'? Where is the evidence of indirect feedback? Does some of the answer to the last two questions come from the data analysis related to Fig 6 and associated text?

Round 2

Reviewer 1 Report

Authors have addressed my comments on the original manuscript; it looks and reads much better. There are a few typos that you might want to correct, e.g.

line 192 experiminets

Table 1, bacteria in italics.

Author contributions needs filling; at  moment it's all still X.X.X.

NCBI Bioproject needs some accession numbers/links to the data.

Pictures of the plants are OK in supplementary; a limitation of the study you might want to address later on is the histopathology at the cellular level. 
